# Role of Social Support in Screening Colonoscopy/Sigmoidoscopy Uptake among U.S. Adults

**DOI:** 10.3390/healthcare12030344

**Published:** 2024-01-30

**Authors:** Benjamin E. Ansa, Biplab Datta, Samah Ibrahim, KM Monirul Islam, Ashley Saucier, Janis Coffin

**Affiliations:** 1Institute of Public and Preventive Health, Augusta University, Augusta, GA 30912, USA; bdatta@augusta.edu (B.D.); saibrahim@augusta.edu (S.I.); kislam@augusta.edu (K.M.I.); 2Department of Health Management, Economics and Policy, Augusta University, Augusta, GA 30912, USA; 3Department of Biostatistics, Data Science and Epidemiology, Augusta University, Augusta, GA 30912, USA; 4Department of Family and Community Medicine, Medical College of Georgia, Augusta University, Augusta, GA 30912, USA; asaucier@augusta.edu (A.S.); jcoffin@augusta.edu (J.C.)

**Keywords:** social support, colorectal cancer, screening, colonoscopy, sigmoidoscopy

## Abstract

Colorectal cancer (CRC) is a major clinical and public health burden. Screening has been shown to be effective in preventing CRC. In 2021, less than 72% of adult Americans had received CRC screening based on the most recent guidelines. This study examined the relationship between social support and screening colonoscopy or sigmoidoscopy uptake among U.S. adults and the socioeconomic factors that impact the relationship. We conducted a cross-sectional study using the 2021 National Health Interview Survey (NHIS) data for 20,008 U.S. adults to assess the weighted rates of screening colonoscopy or sigmoidoscopy among individuals with strong, some, and weak social support. Adjusted binary logistic regression models were utilized to obtain the weighted odds of receiving a screening colonoscopy or sigmoidoscopy among adults with different levels of social support and socioeconomic status. About 58.0% of adults who reported having colonoscopy or sigmoidoscopy had strong social support, compared to 52.0% who had some or weak social support. In addition, compared to adults with weak social support, the weighted adjusted odds of having colonoscopy or sigmoidoscopy were 1.0 (95% C.I. = 0.994, 0.997; *p* < 0.001) and 1.3 (95% C.I. = 1.260, 1.263; *p* < 0.001) for adults with some and strong social support, respectively. Socioeconomic differences were observed in the odds of colonoscopy or sigmoidoscopy uptake based on having strong social support. Having strong social support is an important factor in increasing colonoscopy or sigmoidoscopy screening uptake. Policies and interventions that enhance social support among adults for screening colonoscopy or sigmoidoscopy are warranted.

## 1. Introduction

Colorectal cancer (CRC) is the third most common cancer diagnosed and the second leading cause of cancer-related deaths in both men and women [1]. Timely screening has been proven to be effective in reducing the incidence and mortality rates of CRC [2,3]. The United States Preventive Services Taskforce (USPSTF) concluded with high certainty that screening for CRC in adults aged 50 to 75 years has substantial net benefit [4]. It also concluded with moderate certainty that screening for CRC has a moderate net benefit in adults aged 45 to 49 years and in those aged 76 to 85 years who have never been screened [4]. Screening colonoscopy and sigmoidoscopy reduce CRC mortality rates by 60–75% and 13–50%, respectively [5,6].

Despite the proven advantages of screening, less than 72% of adults aged 50 to 75 years reported being up to date with CRC screening in 2021 [7]. Several factors are associated with screening rates, including age, race/ethnicity, education, insurance coverage, and geographic location [8,9]. Social support is a proven key indicator of preventive healthcare utilization, such as cancer screening [10,11]. It refers to a person’s perception of the availability of help or support from others in their social network [12]. Social support may be defined as a network of family, friends, neighbors, and community members that is available in times of need to give psychological, physical, and financial help [13].

Social support is particularly important in the post-operative management of individuals undertaking colonoscopy and those who choose to have sedation for sigmoidoscopy. Due to the sedation given during the procedure, support in the form of a companion is necessary for driving the individual back home after the procedure. Many healthcare providers do not recommend that patients drive, use power tools, sign legal documents, conduct business, or make important decisions until at least one day after sedation. The Association of periOperative Registered Nurses (AORN) revised its guidelines for monitored sedation and required that pre-operative assessment must include the verification of a caregiver over 18 to drive the patient home [14].

Few published studies have examined the relationship between social networks and CRC screening in specific populations [10,15,16,17] and have reported that social support and interaction are effective in increasing CRC screening rates. These studies, however, targeted specific races and ethnicities, such as African Americans [15] and Latinos [17], and they assessed the relationship between social support and CRC screening in general. They were not specific in the modalities used for screening, such as colonoscopy or stool-based testing. In this study, we assessed the relationship between social support and screening colonoscopy or sigmoidoscopy uptake among the general United States (U.S.) adult population and examined the socioeconomic factors that impact the relationship.

## 2. Materials and Methods

### 2.1. Study Design and Data Source

We conducted a cross-sectional observational study using the 2021 National Health Interview Survey (NHIS) data, a nationally representative household survey managed by the National Center for Health Statistics (NCHS) which is part of the Centers for Disease Control and Prevention (CDC) [18]. Data from the NHIS is used to monitor the health of the U.S. population on a broad range of health topics. Geographically clustered sampling techniques are used to select the sample of dwelling units for the NHIS. The “sample adult” data for individuals aged 18 years or older was used for this study. The total household response rate for the 2021 NHIS sample was 52.8%, and the “sample adult” response rate was 50.9% [19]. NHIS is approved by the Research Ethics Review Board of the National Center for Health Statistics and the U.S. Office of Management and Budget. All NHIS respondents provided oral consent prior to participation [20].

### 2.2. Measures

Outcome variable: Screening colonoscopy or sigmoidoscopy uptake among adults who responded to the survey question “Have you ever had either colonoscopy or sigmoidoscopy?” (Yes/No).

Predictor variable: Social support among adults who responded to the questions “How often do you get the social and emotional support you need? Would you say always, usually, sometimes, rarely, or never?” For this study, we re-coded the responses into strong (always or usually), some (sometimes), and weak (rarely or never) based on a previously published study [21].

Covariates: We included some of the known socioeconomic factors that influence CRC screening rates as covariates for this analysis. They are sex, race, age, marital status, education, health insurance coverage, and income. We used the family income-to-poverty ratio (FIPR) reported in the NHIS as a proxy for annual family income.

### 2.3. Statistical Analysis

We utilized the IBM SPSS version 28.0 (IBM Corp., Armonk, NY, USA) for all statistical analyses. For participants’ characteristics, we used the Chi-square test to assess significant relationships in cross-tabulations between the covariates, the predictor variable (social support), and the outcome variable (colonoscopy or sigmoidoscopy uptake). To produce nationally representative prevalence estimates, we appropriated sample weights to account for the complex survey design of the NHIS. We used binary logistic regression to assess the associations between social support and colonoscopy or sigmoidoscopy uptake adjusting for all the covariates. The less than 50 years age group (40–49) was excluded from the binary logistic regression models. The significance level was set at *p* < 0.05.

## 3. Results

### 3.1. Characteristics of the Study Population

Table 1 shows the unweighted frequencies and percentages of 20,008 adults included in the final analysis, out of which 81%, 11.3%, and 7.7% reported having strong, some, and weak social support, respectively. Most of the study participants were females (55.3%), non-Hispanic White (71.4%), between 60–69 years old (26%), and had a FIPR ≥ 5 (33.7%). The majority were married (51.3%) with greater than a high school education (65.4%) and had health insurance coverage (94.2%).

### 3.2. Prevalence of Colonoscopy or Sigmoidoscopy Uptake

For the unweighted analysis, 61.9% of those who had either a colonoscopy or sigmoidoscopy reported having strong social support. In comparison, among those who reported having some or weak social support, 56.2% and 54.8% had either a colonoscopy or sigmoidoscopy, respectively (Figure 1).

The weighted proportions of adults who reported having colonoscopy or sigmoidoscopy were 57.8%, 52.0%, and 52.1% for those with strong, some, and weak social support, respectively (Table 2). Both the unweighted and weighted analyses show that significant differences based on socioeconomic status were observed in the proportion of respondents who ever had either a colonoscopy or sigmoidoscopy according to the level of social support. Compared to Hispanics, more non-Hispanic Blacks, non-Hispanic Whites, and American Indians and Alaskan Natives (AIANs) reported having strong social support and ever having a colonoscopy or sigmoidoscopy. More respondents who were older, married, had greater than high school education, had health insurance coverage, and had higher FIPR reported having strong social support and ever having a colonoscopy or sigmoidoscopy compared to the corresponding socioeconomic categories.

### 3.3. Adjusted Odds of Screening Colonoscopy or Sigmoidoscopy Uptake

The results of the unweighted binary logistic regression revealed that the adjusted odds of screening colonoscopy or sigmoidoscopy uptake was 1.35 (95% confidence interval (C.I.) = 1.21, 1.49; *p* < 0.001) and 1.06 (C.I. = 0.93, 1.21; *p* = 0.37) for respondents that had strong and some social support, respectively, compared to those who reported having weak social support. These results were like those from the weighted binary logistic regression (Table 3).

Since respondents with strong social support had a significant likelihood of screening colonoscopy or sigmoidoscopy uptake, we used this as the selection variable in the adjusted logistic regression model for assessing the odds of screening colonoscopy or sigmoidoscopy uptake among the socioeconomic groups as covariates (Table 4). The less than 50 years age group (40–49) was excluded from the binary logistic regression models. For the unweighted model, non-Hispanic Blacks and Whites were 1.70 (95% C.I. = 1.41, 2.06; *p* < 0.001) and 1.30 (95% C.I. = 1.12, 1.51; *p* < 0.001) times respectively more likely to report ever having a screening colonoscopy or sigmoidoscopy compared to Hispanics. Non-Hispanic Asians were 0.65 times less likely to report ever having a screening colonoscopy or sigmoidoscopy than Hispanics. Respondents who were 60 years and older, were married, had greater than a high school education, had insurance coverage, and had an FIPR of ≥3 had significantly higher odds of having a screening colonoscopy or sigmoidoscopy compared to the respondents who were less than 60 years old, were not married, had less than a high school education, did not have insurance coverage, and had an FIPR of ≤3. The results of the weighted logistic regression model were like those of the unweighted model.

## 4. Discussion

This study contributes to the literature that multiple factors including strong social support influence CRC screening uptake. Our results reveal that stronger levels of social support have a positive impact on screening colonoscopy or sigmoidoscopy uptake. In addition, the prevalence and odds of screening colonoscopy or sigmoidoscopy uptake were significantly different for the categories of socioeconomic groups. Hispanics and non-Hispanic Asians were less likely to report having strong social support and screening colonoscopy or sigmoidoscopy compared to non-Hispanic Whites and Blacks. Older individuals who were married and had a higher education than high school were more likely to report having strong social support and screening colonoscopy or sigmoidoscopy. Also, those with health insurance coverage and a family income ratio greater than three were more likely to report having strong social support and screening colonoscopy or sigmoidoscopy.

These findings are similar to those from a randomized controlled trial that tested the impact of social support on screening rates among Latino adults in Pennsylvania [17]. A higher return rate of completed useable FIT kit screening tests was observed among participants in the social support arm compared with the control. Participants in the social support arm were 2.67 times as likely to return a completed FIT kit. Another study reported that African Americans with higher social network index scores had higher CRC screening rates [15].

A possible explanation for the positive relationship between social support and CRC screening uptake is that social support is related to screening beliefs, which in turn are related to screening informed decision [22]. When individuals are socially connected and have stable and supportive relationships, they are more likely to make healthy choices and to have better mental and physical health outcomes [23]. Also, supportive social networks may provide individuals with information about screening, the use of screening services, and the reinforcement to use screening services [24]. Another important reason for the positive association between strong social support and CRC screening is that many healthcare providers including the Association of periOperative Registered Nurses (AORN) do require that patients have someone drive them home or accompany them on other transit options to make sure they get home safely after colonoscopy [14]. Individuals without social support who are not provided with alternative means of meeting this requirement may not be able to schedule the screening procedure.

The present study results show that some socioeconomic categories were associated with strong social support. It is imperative for future studies to investigate how these socioeconomic categories access social support to help identify meaningful public health and healthcare interventions that could be targeted at populations at risk of lesser social support.

### Study Limitations

A limitation of this study is that it did not assess the uptake of other valid USPFTF screening modalities for CRC, such as fecal immunochemical testing, high-sensitivity fecal occult blood tests, stool DNA fecal immunochemical testing, and CT colonography [4]. It is important to have evidence-based knowledge of how social support impacts the use of these other modalities, whether this is similar to or different from how colonoscopy/sigmoidoscopy uptake is impacted.

In addition, the NHIS survey question used for this study did not separate colonoscopy from sigmoidoscopy. Since the procedures for colonoscopy and sigmoidoscopy are different, the social support required between the two may not be entirely similar. For the year 2018, the screening rates among U.S. adults for colonoscopy and sigmoidoscopy were 64.3% and 1.9%, respectively. According to this information, it may be correct to assume that most of the responses from the 2021 NHIS survey used for this study were mostly related to colonoscopy rates.

The cross-sectional design of the NHIS survey is another limitation of the study. Having social support and having received a colonoscopy/sigmoidoscopy may have occurred at different time points. For example, someone who is age 70 may be reporting a colonoscopy they received in their fifties, while their report of social support may be what they currently receive. It is possible that they may have had more or less social support at the time of screening than their current level.

## 5. Conclusions

Strong social support is an important factor in increasing colonoscopy or sigmoidoscopy screening uptake. Differences exist among socioeconomic groups in the perception of social support and screening uptake. More research and interventions that promote social support among screening-eligible adults are warranted.

## Figures and Tables

**Figure 1 healthcare-12-00344-f001:**
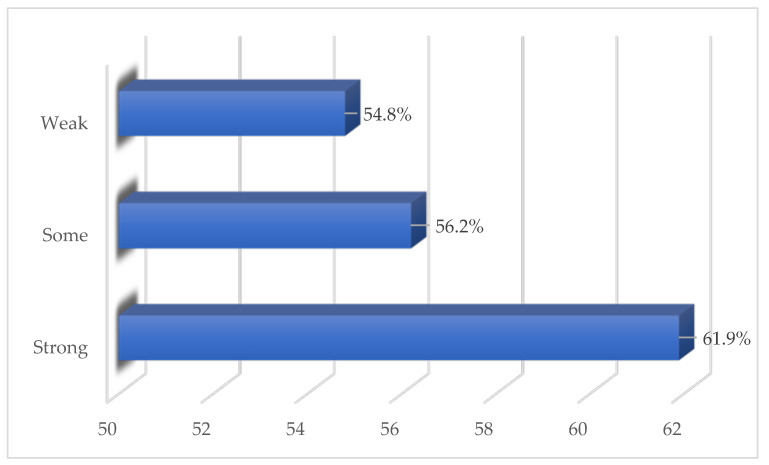
Unweighted proportions of screening colonoscopy or sigmoidoscopy uptake among U.S. adults based on perceived social support: 2021 NHIS Data.

**Table 1 healthcare-12-00344-t001:** Characteristics of U.S. adult respondents to the screening colonoscopy/sigmoidoscopy survey based on the level of social support received: 2021 NHIS data.

Variables Overall	Total N (%) N = 20,008	Strong N = 16,215	Some N = 2257	Weak N = 1536
Sex				
Male	8940 (44.7)	7185 (44.3)	969 (42.9)	786 (51.2)
Female	11,066 (55.3)	9028 (55.7)	1288 (57.1)	750 (48.8)
Refused	2	2	-	-
Race/Ethnicity				
Hispanic	2177 (10.9)	1673 (10.3)	257 (11.4)	247 (16.1)
White, NH	14,289 (71.4)	11,950 (73.7)	1441 (63.8)	898 (58.5)
Black, NH	2090 (10.4)	1552 (9.6)	349 (15.5)	189 (12.3)
Asian, NH	1024 (5.1)	722 (4.5)	156 (6.9)	146 (9.5)
AIAN, NH	115 (0.6)	78 (0.5)	18 (0.8)	19 (1.2)
AIAN, NH + Any other group	148 (0.7)	115 (0.7)	18 (0.8)	15 (1.0)
Others	165 (0.8)	125 (0.8)	18 (0.8)	22 (1.4)
Age (years)				
40–49	4194 (20.9)	3339 (20.6)	540 (23.9)	315 (20.5)
50–59	4604 (23.0)	3641 (22.5)	577 (25.6)	386 (25.1)
60–69	5198 (26.0)	4225 (26.1)	562 (24.9)	411 (26.8)
70–75	2678 (13.4)	2227 (13.7)	265 (11.7)	186 (12.1)
76–85	3334 (16.7)	2783 (17.2)	313 (13.9)	238 (15.5)
Marital status				
Married	10,260 (51.3)	8852 (54.6)	892 (39.5)	516 (33.6)
Living with a partner/unmarried couple	770 (3.8)	634 (3.9)	84 (3.7)	52 (3.4)
Neither	8919 (44.6)	6684 (41.2)	1275 (56.5)	960 (62.5)
Refused	59 (0.3)	45 (0.3)	6 (0.3)	8 (0.5)
Education				
<High School	1567 (7.8)	1114 (6.9)	244 (10.8)	209 (13.6)
High School	5246 (26.2)	4097 (25.3)	664 (29.4)	485 (31.6)
>High School	13,094 (65.4)	10,932 (67.4)	1336 (59.2)	826 (53.8)
Refused	101 (0.5)	72 (0.4)	13 (0.6)	16 (1.0)
Health Insurance Coverage				
Not covered	1124 (5.6)	789 (4.9)	180 (8.0)	155 (10.1)
Covered	18,849 (94.2)	15,398 (95.0)	2074 (91.9)	1377 (89.6)
Refused	35 (0.2)	28 (0.2)	3 (0.1)	4 (0.3)
Income (FIPR)				
0–<1	1789 (8.9)	1189 (7.3)	319 (14.1)	281 (18.3)
1–<2	3401 (17.0)	2513 (15.5)	502 (22.2)	386 (25.1)
2–<3	3219 (16.1)	2528 (15.6)	420 (18.6)	271 (17.6)
3–<4	2477 (12.4)	2045 (12.6)	255 (11.3)	177 (11.5)
4–<5	2376 (11.9)	2010 (12.4)	230 (10.2)	136 (8.9)
≥5	6746 (33.7)	5930 (36.6)	531 (23.5)	285 (18.6)

Note: NH, non-Hispanic; AIAN, American Indian, Alaska Native; FIPR, family income-to-poverty ratio.

**Table 2 healthcare-12-00344-t002:** Proportions of U.S. adult respondents who ever had either a colonoscopy or sigmoidoscopy based on the level of social support received: 2021 NHIS data.

Variables	Unweighted %			Weighted %			*p*-Value
	Strong	Some	Weak	Strong	Some	Weak	
Overall	61.9	56.2	54.8	57.8	52.0	52.1	
Sex							<0.001
Male	61.2	53.3	54.8	56.9	49.0	53.5	
Female	62.6	58.5	54.7	58.6	54.4	50.6	
Race/Ethnicity							<0.001
Hispanic	43.1	43.2	39.3	39.3	38.1	38.8	
White, NH	65.9	61.1	60.5	62.6	57.7	59.4	
Black, NH	62.6	56.7	52.9	58.2	54.2	47.2	
Asian, NH	42.8	35.9	50.7	42.8	35.9	50.3	
AIAN, NH	59.0	50.0	57.9	57.5	38.1	56.0	
Others	41.6	38.9	40.9	49.7	38.2	47.7	
Age (years)							<0.001
40–49	18.1	19.4	18.4	16.8	18.7	19.0	
50–59	56.8	53.9	47.7	56.8	53.9	48.2	
60–69	77.9	71.7	69.3	77.6	68.9	70.3	
70–75	83.3	80.0	71.5	82.6	75.9	70.8	
76–85	79.9	76.0	76.1	79.3	75.5	75.1	
Marital status							
Married	62.7	55.3	55.4	59.1	52.4	52.6	
Living with a partner/unmarried couple	47.9	47.6	40.4	43.9	40.5	43.9	
Neither	62.4	57.6	55.1	57.5	53.2	52.3	
Education							
<High School	53.2	50.0	47.8	47.6	46.1	45.6	
High School	59.8	56.3	49.5	55.0	51.4	47.0	
>High School	63.7	57.3	59.6	60.4	53.6	57.8	
Health Insurance Coverage							<0.001
Not covered	21.4	18.9	18.1	18.2	16.5	16.4	
Covered	64.0	59.5	59.0	60.4	55.9	56.8	
Income (FIPR)							<0.001
0–<1	51.4	49.8	44.5	43.6	44.3	40.8	
1–<2	56.7	52.8	52.1	51.7	47.4	45.9	
2–<3	59.7	55.0	57.2	54.6	51.3	56.2	
3–<4	62.2	58.8	59.9	58.5	54.9	57.5	
4–<5	65.1	63.0	62.5	60.4	55.2	59.8	
≥5	66.1	60.1	59.3	63.3	58.1	59.3	

Note: NH, non-Hispanic; AIAN, American Indian, Alaska Native; FIPR, family income-to-poverty ratio.

**Table 3 healthcare-12-00344-t003:** Adjusted odds of screening colonoscopy or sigmoidoscopy uptake among U.S. adult respondents based on the level of social support received: 2021 NHIS data.

	Unweighted		Weighted	
Variables	Adjusted Odds Ratio (95% Confidence Interval)	*p*-Value	Adjusted Odds Ratio (95% Confidence Interval)	*p*-Value
Social support				
Weak	Reference		Reference	
Some	1.061 (0.932, 1.209)	0.370	0.996 (0.994, 0.997)	<0.001
Strong	1.345 (1.211, 1.494)	<0.001	1.261 (1.260, 1.263)	<0.001

**Table 4 healthcare-12-00344-t004:** Adjusted odds of screening colonoscopy or sigmoidoscopy uptake among U.S. adult respondents based on strong social support.

	Unweighted		Weighted	
Variables	Adjusted Odds Ratio (95% Confidence Interval)	*p*-Value	Adjusted Odds Ratio (95% Confidence Interval)	*p*-Value
Sex				<0.001
Male	Reference		Reference	
Female	1.021 (0.937, 1.112)	0.636	0.997 (0.996, 0.998)	
Race/Ethnicity				<0.001
Hispanic	Reference		Reference	
White, NH	1.302 (1.124, 1.509)	<0.001	1.326 (1.324, 1.328)	
Black, NH	1.703 (1.407, 2.062)	<0.001	1.725 (1.721, 1.728)	
Asian, NH	0.653 (0.516, 0.827)	<0.001	0.676 (0.674, 0.678)	
AIAN, NH	1.333 (0.762, 2.331)	0.313	1.797 (1.785, 1.808)	
Others	0.752 (0.438, 1.291)	0.301	0.634 (0.630, 0.638)	
Age (years)				<0.001
50–59	Reference		Reference	
60–69	2.804 (2.529, 3.109)	<0.001	2.759 (2.755, 2.762)	
70–75	3.889 (3.396, 4.453)	<0.001	3.734 (3.728, 3.740)	
76–85	3.559 (3.142, 4.033)	<0.001	3.418 (3.413, 3.423)	
Marital status				<0.001
Married	Reference		Reference	
Living with a partner/unmarried couple	0.698 (0.558, 0.873)	0.002	0.737 (0.736, 0.739)	
Neither	0.717 (0.654, 0.785)	<0.001	0.716 (0.715, 0.716)	
Education				<0.001
<High School	Reference		Reference	
High School	1.160 (0.981, 1.372)	0.082	1.023 (1.021, 1.025)	
>High School	1.409 (1.192, 1.665)	<0.001	1.328 (1.325, 1.330)	
Health Insurance Coverage				<0.001
Not covered	Reference		Reference	
Covered	3.393 (2.729, 4.218)	<0.001	3.604 (3.595, 3.612)	
Income (FIPR)				<0.001
0–<1	Reference		Reference	
1–<2	0.999 (0.842, 1.186)	0.993	1.159 (1.157, 1.162)	
2–<3	1.210 (1.014, 1.444)	0.034	1.369 (1.366, 1.372)	
3–<4	1.440 (1.190, 1.742)	<0.001	1.727 (1.724, 1.731)	
4–<5	1.837 (1.509, 2.236)	<0.001	1.967 (1.962, 1.971)	
≥5	2.007 (1.683, 2.392)	<0.001	2.229 (2.224, 2.233)	

Note: FIPR, family income-to-poverty ratio. Others, other single and multiple races.

## Data Availability

NHIS datasets that were analyzed for this study may be accessed at https://www.cdc.gov/nchs/nhis/2021nhis.htm (accessed on 14 September 2023).

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
