# Peer review of "Role of Social Support in Screening Colonoscopy/Sigmoidoscopy Uptake among U.S. Adults"

_healthcare, 2024, doi:10.3390/healthcare12030344_

Round 1

Reviewer 1 Report

Comments and Suggestions for Authors

This article is on a topic that will likely be of interest to readers, and it provides more evidence on the need to design effective interventions to improve social support. I have some comments that may improve this article.

Introduction: You may want to give some more rationale for focusing on these two tests specifically. One reason could be the need to have someone accompany the patient when the procedure is performed. 

Methods: I would suggest justifying showing both unweighted and weighted results. Generally, it isn't necessary to show the unweighted results because the weighted estimates are reflective of the U.S. population, which the survey was designed to provide. More information can be found here: https://www.cdc.gov/nchs/nhis/data-questionnaires-documentation.htm

Table 1. For age, you may want to include the minimum age for the youngest age category (<50 years). Also, there's a typo with the less than high school category (should be "<"). Per U.S. OMB standards on reporting race and ethnicity, AI/AN stands for "American Indian/Alaska Native." 

Table 4. For consistency, you may only want to report to two decimal places with the odds ratios presented. 

Discussion: There are also logistical considerations with colonoscopy that you may want to address as well. If a person has weak social support, they may not have anyone to rely on to accompany them to the procedure. Many clinics require patients to have someone drive them home, or accompany them on other transit options to make sure they get home safely. 

Another limitation is the cross sectional design of this survey. I recommend noting as a limitation that social support and having received colonoscopy may have occurred at different time points. For example, someone who is age 70 may be reporting a colonoscopy they received in their fifties, while their report of social support may be what they currently receive. It's possible they may have had more or less social support at the time of screening than their current level. 

Data availability statement: Replace BRFSS with NHIS.

Author Response

Thank you for your precious time and effort in critically reviewing our manuscript and your valuable comments are highly appreciated. We have tried to revise the manuscript accordingly and believe that your comments have helped to further improve the quality of our paper.

I have some comments that may improve this article.

  • Introduction: You may want to give some more rationale for focusing on these two tests specifically. One reason could be the need to have someone accompany the patient when the procedure is performed.

Author Response: Thank you very much for this suggestion. We have included a paragraph and a reference in the introduction accordingly.

Social support is particularly important in the post-operative management of individuals undertaking colonoscopy/sigmoidoscopy. Due to the sedation given during the procedure, support in the form of a companion is necessary for driving the individual back home after the procedure. Many healthcare providers do not recommend that patients drive, use power tools, sign legal documents, conduct business, or make important decisions until at least one day after sedation. The Association of periOperative Registered Nurses (AORN) revised their guidelines for monitored sedation and required that pre-operative assessment must include verification of a caregiver over 18 to drive the patient home”.

  • Methods: I would suggest justifying showing both unweighted and weighted results. Generally, it isn't necessary to show the unweighted results because the weighted estimates are reflective of the U.S. population, which the survey was designed to provide. More information can be found here: https://www.cdc.gov/nchs/nhis/data-questionnaires-documentation.htm.

Author Response: We agree with you that the weighted estimates are reflective of the U.S. population. We have reported both the weighted and unweighted estimates to show how results from the sample size compare with the general population. It is optional to report the unweighted results as NHIS does not mandate that the unweighted results may not be reported.

  • Table 1. For age, you may want to include the minimum age for the youngest age category (<50 years). Also, there's a typo with the less than high school category (should be "<"). Per U.S. OMB standards on reporting race and ethnicity, AI/AN stands for "American Indian/Alaska Native."

Author Response: Thank you for the observation and suggestion. We have included the minimum age: “40-49”. Also, thank you for picking up the error in AI/AN and “>” for education. These have been corrected accordingly.

  • Table 4. For consistency, you may only want to report to two decimal places with the odds ratios presented.

Author Response: Thank you for this observation. We were mindful of the need for being consistent in reporting two decimal places with the odds ratios, however, this did not make much sense with the weighted results. For example, in table 3, we would have had OR=1.26 (1.26, 1.26) if we reported to two decimal places, and this will no longer be a real confidence interval. We have therefore reported the 3 decimal places for consistency.

  • Discussion: There are also logistical considerations with colonoscopy that you may want to address as well. If a person has weak social support, they may not have anyone to rely on to accompany them to the procedure. Many clinics require patients to have someone drive them home or accompany them on other transit options to make sure they get home safely.

Author Response: Thank you for this important suggestion. We have included a couple of sentences in the discussion to reflect this: “Another important reason for the positive association between social support and CRC screening is that many healthcare providers including the Association of periOperative Registered Nurses (AORN) do require that patients have someone drive them home or accompany them on other transit options to make sure they get home safely after colonoscopy or sigmoidoscopy. Individuals without social support that are not provided with alternative means of meeting this requirement may not be able to schedule the screening procedure”.

  • Another limitation is the cross-sectional design of this survey. I recommend noting as a limitation that social support and having received colonoscopy may have occurred at different time points. For example, someone who is age 70 may be reporting a colonoscopy they received in their fifties, while their report of social support may be what they currently receive. It's possible they may have had more or less social support at the time of screening than their current level.

Author Response: Thank you for raising this point. We have borrowed from your valuable comment and included this in the discussion: “Another limitation is the cross-sectional design of the NHIS survey. Having social support and having received a colonoscopy/sigmoidoscopy may have occurred at different time points. For example, someone who is age 70 may be reporting a colonoscopy they received in their fifties, while their report of social support may be what they currently receive. It is possible they may have had more or less social support at the time of screening than their current level.”

Data availability statement: Replace BRFSS with NHIS.

Author Response: Thank you once again for catching this error. Your attention to details is highly appreciated. We have revised the manuscript to reflect this.

Reviewer 2 Report

Comments and Suggestions for Authors

Colorectal cancer is one of the top 3 most commonly seen cancer among adults and the need for screening is huge. This article shows very good insight into the social variable affecting the person to have colonoscopy. This is a review of data retrospective is the only limitation.  Otherwise results are well populated and overall article gives out good message that social support plays a vital role in increasing the colonoscopy screening numbers. 

Author Response

Reviewer 2 comment:

Colorectal cancer is one of the top 3 most commonly seen cancer among adults and the need for screening is huge. This article shows very good insight into the social variable affecting the person to have colonoscopy. This is a review of data retrospective is the only limitation.  Otherwise results are well populated and overall article gives out good message that social support plays a vital role in increasing the colonoscopy screening numbers.

Author Response:

Thank you for your precious time and effort in reviewing our manuscript. We appreciate your kind and encouraging comments.

Reviewer 3 Report

Comments and Suggestions for Authors

This study seeks to use the data collected from the 2021 NHIS to study the reason behind the poor adherence to colonoscopy guidelines in the United States. On the specifics of the study, I had the following comments

1. It needs to be addressed why the survey population does not reflect the US population by ethnic demographics.

2. it is not clear why Hispanics were made the reference in odds ratio evaluation among those with strong social support. 

3. Colonoscopy and sigmoidoscopy carries significantly different social challenges both in terms of the prep time required and the need for post procedure recovery. As a result, they should not be lumped together in an evaluation for whether or not social support had a role in its adherence.

On the study as a whole, I found though the results were based on solid statistical work, that it does not account for some glaring shortcoming of the original data set:

1. As the authors acknowledges already, the NHIS does not take in to account the other valid and used methods for CRC screening outlined by the USPFTF. It is difficult to know if the patients not reporting an endoscopic examination is because of the suggested reasons of social support, or that a significant subset of the data comes from a group that was preferentially screened through another valid modality.

2. Additionally, over 20% of the survey comes from patients younger than the age of 50. While the current guidelines have decreased to 45 as the recommended time for initial screening colonoscopy, the people in this survey likely reflect those who have previously been seen by their healthcare provider prior to this guideline change and would not be offered a colonoscopy in the first place. It is my opinion that all patients under the age of 50 should be excluded in the analysis in the first place.  

Given these concerns for the data set, it is hard to support this was the correct use of NHIS. 

Author Response

Dear Reviewer,

We thank you for your precious time and effort in reviewing our manuscript and your valuable comments are highly appreciated. Below are the author's responses to your comments.

This study seeks to use the data collected from the 2021 NHIS to study the reason behind the poor adherence to colonoscopy guidelines in the United States. On the specifics of the study, I had the following comments:

  1. It needs to be addressed why the survey population does not reflect the US population by ethnic demographics.

Author Response: Thank you for your observation and comment. This point has already been addressed in the Statistical Analysis subsection of the Methodology: “To produce nationally representative prevalence estimates, we appropriated sample weights to account for the complex survey design of the NHIS.” One cannot predict the number and ethnicity of respondents to a survey and therefore, the NHIS has provided a statistical tool called “weighting” for adjustment so that the results may be generalizable and reflect the US population.

  1. It is not clear why Hispanics were made the reference in odds ratio evaluation among those with strong social support.

Author Response: Thank you for your comment. In statistics, there is no rule that states which member of a variable category must be used as the reference in analyzing odds ratios. The decision is left for the analyst based on their judgement. Choosing Hispanics was simply the choice of the authors.

  1. Colonoscopy and sigmoidoscopy carry significantly different social challenges both in terms of the prep time required and the need for post procedure recovery. As a result, they should not be lumped together in an evaluation for whether or not social support had a role in its adherence.

Author Response: Thank you for your comment. For both colonoscopy and sigmoidoscopy many healthcare providers require that the patient must have somebody (social support) to accompany them home post-op. We have added a few sentences in the introduction and discussion sections of the revised manuscript to reflect this. This study is not about the prep time or post-op procedure, but it is about the availability of support and its association with colonoscopy and sigmoidoscopy screening uptake. We agree with the owners of the NHIS data that the two procedures may be joined together for this purpose.

On the study as a whole, I found though the results were based on solid statistical work, that it does not account for some glaring shortcoming of the original data set:

  1. As the authors acknowledge already, the NHIS does not take into account the other valid and used methods for CRC screening outlined by the USPFTF. It is difficult to know if the patients not reporting an endoscopic examination is because of the suggested reasons of social support, or that a significant subset of the data comes from a group that was preferentially screened through another valid modality.

Author Response: Thank you for your comment. This study aimed at only including the participants that reported having a colonoscopy or sigmoidoscopy based on whether they had social support or not. Therefore, the significant subset of the data coming from a group that was preferentially screened through other valid modality does not affect the presented result. In fact, one of our future projects is to assess the relationship between social support and screening uptake with the other modalities.

  1. Additionally, over 20% of the survey comes from patients younger than the age of 50. While the current guidelines have decreased to 45 as the recommended time for initial screening colonoscopy, the people in this survey likely reflect those who have previously been seen by their healthcare provider prior to this guideline change and would not be offered a colonoscopy in the first place. It is my opinion that all patients under the age of 50 should be excluded in the analysis in the first place.

Author Response: Thank you for your comment. We do not agree that all patients under the age of 50 should be excluded in the analysis. Prior to the 2021 change in the USPSTF recommendations for CRC screening, previous USPSTF recommendations were that younger people (less than 50 years) with high risk for CRC may be screened. It is therefore okay to include this population in our analysis.

  1. Given these concerns for the data set, it is hard to support this was the correct use of NHIS.

Author Response: Thank you for your comment. We have a different opinion and we believe that this is indeed an appropriate use of the NHIS data.

Round 2

Reviewer 3 Report

Comments and Suggestions for Authors

Thank you for your responses. Here are my comments:

1. "Thank you for your comment. For both colonoscopy and sigmoidoscopy many healthcare providers require that the patient must have somebody (social support) to accompany them home post-op. We have added a few sentences in the introduction and discussion sections of the revised manuscript to reflect this. This study is not about the prep time or post-op procedure, but it is about the availability of support and its association with colonoscopy and sigmoidoscopy screening uptake. We agree with the owners of the NHIS data that the two procedures may be joined together for this purpose."

Reviewer Reply: This is not true. Flexible sigmoidoscopy does not routinely require sedation and thus does not always require someone to accompany them. In many institutions including large academical centers (i.e. Mayo Clinic) in the United States, flexible sigmoidoscopies are offered without sedation. Unless it can be identified which patients went to a clinical practice that required sedation for a flex sig, all the flex sig patients needs to be excluded. 

2. "Thank you for your comment. This study aimed at only including the participants that reported having a colonoscopy or sigmoidoscopy based on whether they had social support or not. Therefore, the significant subset of the data coming from a group that was preferentially screened through other valid modality does not affect the presented result. In fact, one of our future projects is to assess the relationship between social support and screening uptake with the other modalities."

Reviewer Reply: This study reports a percentage of patients who do not get endoscopic screening. The question is "How do we know if these patients didn't get endoscopic screening because they already had another form of CRS screening?" Because if that was the case, their decision may be independent of social support. Hence this is not accounted for in the study. 

3. "Thank you for your comment. We do not agree that all patients under the age of 50 should be excluded in the analysis. Prior to the 2021 change in the USPSTF recommendations for CRC screening, previous USPSTF recommendations were that younger people (less than 50 years) with high risk for CRC may be screened. It is therefore okay to include this population in our analysis." 

Reviewer reply: As the author states, young people "with high risk for crc" may be screened if they are less than 50. Nearly all patients under the age of 50 would NOT fit into this category and yet is included in the statistical analysis. Either you are able to pre-select all high risk patients under the age of 50 in your data set to analyze, or these patients needs to be excluded. 

Difference in opinion on what is appropriate use of this data set aside, it is difficult to understand why requests to refine data set prior to analysis is not possible to be accommodated. 

Author Response

Dear reviewer,

We thank you again for your time and commitment to this review process. We have taken great steps in ensuring that we address your comments to the best of our ability.

  1. Reviewer Comment

This is not true. Flexible sigmoidoscopy does not routinely require sedation and thus does not always require someone to accompany them. In many institutions including large academical centers (i.e. Mayo Clinic) in the United States, flexible sigmoidoscopies are offered without sedation. Unless it can be identified which patients went to a clinical practice that required sedation for a flex sig, all the flex sig patients needs to be excluded.

Author Response

As stated in the first round of revision, the focus of this study is not about the differences in procedures between colonoscopy and sigmoidoscopy. However, to address your point, we have done the following:

In lines 213-214 under Study limitations, we included the statement “Since the procedures for colonoscopy and sigmoidoscopy are different, the social support required between the two may not be entirely similar”.

  1. Reviewer Comment: 

This study reports a percentage of patients who do not get endoscopic screening. The question is "How do we know if these patients didn't get endoscopic screening because they already had another form of CRS screening?" Because if that was the case, their decision may be independent of social support. Hence this is not accounted for in the study. 

Author Response

Again, the focus of this study is to assess the relationship between social support and colonoscopy/sigmoidoscopy uptake for CRC screening. In table 1, we described the total population that responded to the NHIS survey question used for this study.

The main results relevant to answering our research question are in tables 2, 3 and 4, where the population is restricted to only those that received a colonoscopy/sigmoidoscopy.

  1. Reviewer Comment

As the author states, young people "with high risk for crc" may be screened if they are less than 50. Nearly all patients under the age of 50 would NOT fit into this category and yet is included in the statistical analysis. Either you are able to pre-select all high risk patients under the age of 50 in your data set to analyze, or these patients needs to be excluded.

Difference in opinion on what is appropriate use of this data set aside, it is difficult to understand why requests to refine data set prior to analysis is not possible to be accommodated.

Author Response

In the highlighted texts in lines 105-106, we included the statement “The less than 50 years age group (40-49) was excluded from the binary logistic regression models”.

Though the revised results were not much different from the initial results from the first model that included the less than 50 age group, we decided to replace the initial results with the revised results in the narrative and table 4. See highlighted texts in lines 152-163 and table 4.
